# Incidence of Sport Injuries in the Manchester 2019 World Taekwondo Championships: A Prospective Study of 936 Athletes from 145 Countries

**DOI:** 10.3390/ijerph20031978

**Published:** 2023-01-20

**Authors:** Hee Seong Jeong, Dae Hyoun Jeong, David M. O’Sullivan, Hyung-Pil Jun, Min Jin Kim, Inje Lee, Hyung Gyu Jeon, Sae Yong Lee

**Affiliations:** 1Department of Sports and Health Management, Mokwon University, Daejeon 35349, Republic of Korea; 2International Olympic Committee Research Centre Korea, Seoul 03722, Republic of Korea; 3Department of Family and Community Medicine, School of Medicine, University of Maryland, Baltimore, MD 21201, USA; 4Division of Sports Science, Pusan National University, Busan 46241, Republic of Korea; 5Department of Physical Education, Dong-A University, Busan 49315, Republic of Korea; 6Department of Physical Education, Yonsei University, Seoul 03722, Republic of Korea; 7Department of Sports Rehabilitation Medicine, Kyungil University, Gyeongsan 38428, Republic of Korea; 8Institute of Convergence Science, Yonsei University, Seoul 03722, Republic of Korea

**Keywords:** epidemiology, injury prevention, Taekwondo injury, incidence, combat sports

## Abstract

We aimed to describe injury incidence and patterns at the 2019 World Taekwondo Championships (WTC), and to compare them with those of previous WTCs, based on new World Taekwondo (WT) competition rules, medical codes, and the Protector and Scoring System (PSS). This prospective cohort study utilized data obtained through the WT Injury Surveillance System. All athletes with injuries were evaluated by on-site sports medicine specialists, and ultrasonography was used to assess all musculoskeletal injuries. Of 936 athletes, 60 injuries were recorded (6.4 injuries/100 athletes, 95% confidence intervals [CI]: 4.8–8.0), and 4.5% (n = 42) sustained at least one injury. Males had a higher risk of sustaining injuries than females (incidence rate ratio: 1.57; 95% CI: 0.89–2.76). The most common sites, type, and mechanism were lower extremities (n = 26, 43.33%), contusion/hematoma/bruises (n = 33, 55.0%), and contact with another athlete (n = 50, 83.33%). Overall, the injury patterns associated with the mechanism of injury were similar in both the 2019 and 2017 WTCs. Refined WT competition rules and a re-established PSS at the 2019 WTC reduced the overall and severe injury incidence. Our findings can help revise Taekwondo competition rules, enhance protective equipment, optimize on-site venue medical systems, and develop injury prevention projects.

## 1. Introduction

Taekwondo Sparring is an official combat sport in the Olympics [1]. Of the numerous international Taekwondo competitions hosted every year, the quadrennial Summer Olympic Games hosted by the International Olympic Committee (IOC) and the biannual World Taekwondo Championships (WTC) hosted by World Taekwondo (WT) are regarded as two of the largest international Taekwondo events [2,3]. Taekwondo has shown a high injury incidence rate due to the nature of martial art-based full-contact sports. Taekwondo was reported to have the highest injury rate in London in 2012 [4], the second highest in Beijing in 2008 [5], and the fourth highest in Rio in 2016 [6], urging the IOC and WT to address the safety concerns of Taekwondo. In addition, it was revealed that the injured athlete rates at the 1999 Edmonton WTC [7] and the 2017 Muju WTC were 19.5% and 13.5%, respectively. Since the 2016 Rio Olympic Games (Rio 2016), WT has introduced amended competition rules and a precise protector and scoring system (PSS) [8]. These changes were also applied at the Muju 2017 WTC [9] and Manchester 2019 WTC promoted by WT [10].

The IOC conducts sports epidemiology studies at international sporting events, including the Olympics, to analyze risk factors and mechanism of injuries and illnesses for health promotion and injury and illness prevention among athletes [11]. The IOC Medical Commission initiated the IOC Injury Surveillance System (ISS) at the 2004 Athens Olympics, which was improved to be applied to individual and team sports at the 2008 Beijing Olympics [5] and further improved at the 2010 Vancouver Olympics to be able to collect both injury and illness data from the 2012 London Olympics [4], 2014 Sochi [12], 2016 Rio [13], 2018 Pyeongchang [14], and 2020 Tokyo Olympics [15]. Therefore, WT signed a Memorandum of Understanding (MoU) with the IOC Research Center KOREA (representative institution: Yonsei Institute of Sports Science and Exercise Medicine, YISSEM) in 2017 to comply with the IOC’s policy, and from the 2017 Muju WTC, WT has applied the ISS to major competitions hosted by WT, organized a committee to prevent injuries and improve performance of players, and have research and education systems in place [16].

Despite many efforts, injury surveillance reports and other epidemiologic studies conducted by the Olympics and international institutions have some systemic barriers, including poor understanding and compliance with the injury monitoring system of venue medical staff, team doctors, coaches, etc., to collect injury data, inaccurate injury reports relying on the self-reporting of athletes in the field of play, unreported athlete-exposures during the competition, and inconsistent medical codes for each sport [3,17]. However, there is a lack of professional and systematic education on the system or approach to addressing these systemic problems [3]. Therefore, WT developed a new medical code in 2017 to address these problems and applied it to the 2017 Muju WTC and 2019 Manchester WTC. The new medical code includes a web-based Injury Surveillance System (ISS), utilization of on-site ultrasound, and medical staff education.

In this study, we aimed to analyze the incidence and patterns that occurred during the 2019 Manchester WTC based on the new WT competition rules and medical codes in comparison to previous WTCs. In addition, the ISS applied in this study is paramount to developing an injury prevention program based on evidence from the large data set that will contribute to the future development of WT competition rules, improve the PSS, and enhance venue medical services suitable for Taekwondo competitions. At the 2019 WTC, a PSS was used that included an electronic headgear and chest protector for full automation of the scoring system, which was the same as that used in the 2017 WTC [3].

## 2. Methods

We conducted a prospective cohort study using our modified ISS, which is more suitable for Taekwondo than the IOC’s injury and illness surveillance system [13], which was implemented during the 2019 WTC, held for 5 days from 15 to 19 May 2019, in Manchester, United Kingdom [10]. To proceed with the same injury surveillance process as at the 2017 WTC, we held a full-day workshop prior to the competition to train team doctors and team trainers from all participating countries and organizing committee venue medical staff to accurately use the online injury and illness surveillance systems and the on-site portable ultrasound for more accurate on-site diagnosis and injury surveillance. We obtained approval from WT and the Yonsei University Institutional Review Board, which declared that it was ethical to collect WT athletes’ injury and illness data for this study (IRB no. 7001988-201708-HR-245-04). In addition, we received approval from the Mokwon University Institutional Review Board for collecting the injury profiles of Taekwondo athletes and conducting ongoing research. (IRB no. 2022-002). Moreover, we declare that the study design, methodology, structure, and flow of the manuscript, and tables and illustrations of this article, were written in a way similar to our previous study (HS Jeong et al., 2017 WTC) for easier comparison by readers [3].

### 2.1. Participants

All Taekwondo athletes who participated in the 2019 WTC were enrolled in this study. A total of 936 athletes from 145 countries and one refugee team participated in the 2019 WTC. Of the 960 scheduled games, the remaining 53 matches did not happen either because the athletes either failed the weigh-in procedure or withdrew from the competition. Athletes participated in the Taekwondo sparring competition in the WTC comprising three rounds of 2 min each with a 1 min break [10].

### 2.2. Data Collection and Implementation

Researchers collected data using the following methods.

(1)All the national team physicians, athletic trainers, and venue medical staff were requested to report the occurrence of new injuries daily either by using a paper form or an online database system (iociss.com, previously known as wtfiss.com, accessed in 2022).(2)Authors collected injury data from venue medical staff of the Organizing Committee supervised by WT medical officers at the venue medical center and/or from specialists at tertiary hospitals designated by the Organizing Committee of the 2019 Manchester WTC who treated athletes with any acute injury or illness.(3)In a total of seven Taekwondo courts, researchers assigned six medical staff to each court (one sports medicine specialist (MD), one athletic trainer (AT), one registered nurse (RN), and three paramedics) to closely monitor matches and record the mechanism of injury and injured area for the Taekwondo-related injury profiles.(4)Authors used two portable ultrasound machines (Mindray M7 ver. 2015, Mahwah, NJ, USA) with a light-frequency linear transducer (L14-6s) and convex transducer (C5-2s) in the venue medical center for thorough evaluation and more accurate point-of-care diagnosis of any musculoskeletal injury.(5)Researchers recorded any injury data on-site in the venue and from the medical staff at a designated hospital to which the injured or ill athletes were directly transferred emergently during the match. The medical staff signed the medical information release form for all injured athletes.(6)After the competition, we corrected or updated injury data recorded on-site in the venue by retrospective video review of each match in which the injury occurred.

### 2.3. Injury Report Form

A paper and an online database form similar to those used in Rio 2016 and 2017 Muju WTC were used for the injury report [3,18]. Each injury was regarded as new (pre-existing, not fully rehabilitated conditions were not recorded) or recurring (athletes who had returned to full participation after recovery from a previous condition) musculoskeletal problems, concussions, or other medical conditions that occurred during the competition/training at 2019 WTC that necessitated medical attention regardless of the reason [13]. Regardless of the severity of the injury, all injuries were reported through the Injury report form. Injury severity was defined as the time lost from training/competition of <1, 1–3, 4–7 days, or >7 days [3,18,19]. In cases of multiple injury types, only the most severe diagnosis was chosen per injury for analysis [13,18].

### 2.4. Statistical Analyses

All statistical analyses were conducted using SPSS version 24.0 (IBM Corp., Armonk, NY, USA) and Microsoft Excel (Microsoft Corp., Redmond, WA, USA). Confidence intervals of 95% were considered as statistically significant. Taekwondo-related injury variables were assessed using univariate analyses and descriptive statistics. The injury incidence rate (IR) was measured using the overall competition and/or training, with clinical incidence as injuries per 100 athletes during WTC 2019 [3,6]. The number of injuries was calculated per 1000 athlete-days, in which athlete-days represented the total number of athletes multiplied by 5 days. IR of competition was expressed as the number of injuries per 1000 athlete-exposures (AEs; 1 exposure is one athlete participating in one match) and per 1000 min-exposures (MEs; 1 min of exposure is one athlete participating in a match for 1 min) [20,21]. We calculated the total game minutes using only the actual fight time [20]. We also measured the average injury risk for one athlete per 1000 AEs using the following formula [20,21]:IR/1000AEs = Total number of injuries/(number of athletes × number of total game matches) × 1000.

The average injury risk for one athlete per 1000 MEs was calculated using the following formula [20,21]:IR/1000MEs = Total number of injuries/(number of athletes × number of total game minutes) × 1000

The incidence rate ratio (IRR) and 95% confidence intervals (CI) were calculated to measure the strength of the associations between male and female athletes using a Poisson model, assuming a constant hazard per sex [21].

## 3. Results

A total of 936 athletes participated in the 2019 Manchester WTC (Table 1). Of these athletes, 544 were male (58.10%) and 392 were female (41.89%), with an average age of 22.0 years (22.50 and 21.40 years, respectively). Throughout the 5 days of the 2019 WTC, 907 matches (male: 533, female: 374) and 3932 minutes occurred (male: 2127, female: 1805). The weight divisions in which the highest number of athletes participated were the <68 kg (n = 86) division for male athletes and the <57 kg (n = 63) division for female athletes.

Table 1 is presented in a similar format to the table from 2017 WTC MUJU, HS Jeong [3] for easier comparison by readers.

### 3.1. Overall Incidence of Injury

A total of 60 injuries were recorded, with an overall clinical incidence of 6.41 injuries per 100 athletes (95% CI: 4.79–8.03), which corresponds to 12.82 injuries per 1000 athlete-days (95% CI: 9.58–16.06); 4.50% (n = 42) of all athletes who participated in the 2019 WTC experienced at least one injury. Among all athletes with injuries, 22 (36.67%) with suspected serious injuries were transferred to designated hospitals for further evaluation and management after a primary assessment in the venue medical center.

During the competition, we observed 4680 athlete-days, 1814 athlete-exposures (AEs), and 7864 min-exposures (MEs). Injuries per 1000 AEs was 33.08 (95% CI: 24.71–41.45), presenting significantly higher rates in male athletes (38.46/1000 AEs) than in female athletes (25.40/1000 AEs). The total injuries per 1000 MEs to game minutes were 7.63 (95% CI: 5.70–9.56), and higher rates were observed in male athletes (9.64/1000 MEs) than in female athletes (5.26/1000 MEs); male athletes were at a higher risk of experiencing injuries during competition than female athletes (IRR 1.57; 95% CI: 0.89–2.76) (Table 1).

Overall, athletes tended to sustain injuries during rounds 2 and 3 of a three-round match (n = 19, 31.70%). In particular, for male athletes, a high number of injuries (n = 14, 34.10%) were sustained during a sudden death match (extended round after tied result in the third round). Furthermore, for male athletes, the weight division with the highest number of injuries was the <68 kg division (n = 9, 21.95%), but the highest injury rate per 1000 AEs was in the >87 kg division (70.0/1000 AEs). For female athletes, the weight division with the highest number of injuries was the <49 kg division (n = 6, 31.57%), but the highest injury rate per 1000 AEs was in the <46 kg division (55.60/1000 AEs).

### 3.2. Site and Cause of Injury Based on Diagnosis

The lower extremities (n = 26, 43.33%) were the most commonly injured areas, with the thigh and knee (n = 6, 10.0% of total injuries) being the most frequently injured followed by the groin (n = 5, 8.33%). In the head and trunk (n = 23, 38.33%), the face was the most frequently injured site (n = 15, 25.0%), followed by the head and ribs (n = 3, 5.0%). Additionally, in the upper extremities (n = 11, 18.33%), the hand was the most frequently injured site (n = 7, 11.67%) (Table 2).

Contusion/hematoma/bruises were the most common type of injury (n = 33, 55.0%), followed by ligament rupture/sprain (n = 7, 11.67%) and fracture (n = 6, 10.0%) (Table 2). The face was the most common site of contusion/hematoma/bruise, and this occurred primarily by contact with the opponent during offensive/defensive moves. The most common sites of ligament rupture/sprain were the knee and ankle, and these occurred primarily due to non-contact reasons. The hand was the most common fracture site. Of the three concussions, one was classified as mild. Fortunately, the two other athletes who were transported to the hospital did not experience time loss from the competition (Table 2).

### 3.3. Severity of Injury

Of all the injuries, 8.33% (n = 5) were estimated to result in no time loss from competition or training. It was estimated that 55.0% (n = 33) of the injuries would result in an absence from competition or training of 1–3 days; in 21.67% (n = 13) of injuries, time loss was of 4–7 days; time loss of >7 days was seen in 15.0% (n = 9) of cases. Additionally, 36.67% (n = 22) of the injuries were classified as severe, with an estimated time loss of ≥4 days from competition or training. These injuries comprised seven ligament ruptures/sprains, six fractures, three concussions, three dislocation/subluxations, and three muscle and tendon rupture/strain/tendinosis.

### 3.4. Mechanism of Injury

We observed that the most common mechanism of injury was contact with another athlete (total n = 50, 83.33%), followed by non-contact (total n = 6, 10.0%), which included injuries from footwork during offensive/defensive moves and avoiding kicking/punching with another athlete. This was followed by recurrence of previous injury corresponding to overuse (total n = 4, 6.67%) (Figure 1). Interestingly, the injury pattern related to the mechanism of injury was similar in both the 2019 and 2017 WTCs. That is, male athletes were more prone to contact injuries from their opponents, whereas females were prone to non-contact injuries.

## 4. Discussion

We aimed to analyze injuries that occurred at the Manchester 2019 WTC, in which newly amended WT competition rules and medical codes were applied, and compare the injury profile in the 2019 WTC with that of previous WTCs to generate more scientific evidence for developing Taekwondo competition rules, improving the venue medical system, and designing an injury prevention program. In this prospective study, we would like to discuss the results of our prospective epidemiologic study that overcame the limitations of previous studies that were mainly attributable to on-site medical staff’s poor understanding of and non-compliance with the ISS, unclear recording of injury diagnosis and medical code, and overlooking exposure time. Our study revealed the following: (1) 60 injuries occurred in 936 Taekwondo athletes during the Manchester 2019 WTC; (2) the most common site of injury was the lower extremities (n = 26, 43.33%); (3) the most common injury types were contusions, hematomas, and bruises (n = 33; 55.0%); (4) most injuries resulted in time loss from training or competition of 1–3 days; (5) most injuries were caused by contact with opponents (n = 50, 83.33%); and (6) the overall injury incidence rate was lower than those observed in previous Olympic games or WTCs, but injuries caused by contact with the opponent were actually higher.

### 4.1. Injuries in WTC

We noted that 6.41% of all athletes sustained musculoskeletal injuries during the 2019 WTC (33.08/1000 AEs), with higher rates in male athletes (n = 41, 38.46/1000 AEs) than in female athletes (n = 19, 25.40/1000 AEs). Male athletes were at a higher risk of injury during the competition than female athletes (IRR 1.57; 95% CI: 0.89, 2.76).

The incidence rate at Manchester 2019 was lower than that in previous reports that were held before the introduction of the new PSS: 34 of 126 athletes at the Beijing 2008 Summer Olympic Games (27/100 clinical incidence) [5], 50 of 128 athletes at the London 2012 Summer Olympic Games (39/100 clinical incidence) [4], and 30 of 127 athletes at the Rio 2016 Olympic Summer Games sustained injuries (24/100 clinical incidence) [6]. At the Muju 2017 WTC, in which Taekwondo was the only sport discipline, 131 of 971 athletes experienced injuries (13.5%) [3]. In the 2008, 2012, and 2016 Olympics, the PSS did not include electronic headgear for automated scoring systems. However, since the 2017 WTC, the PSS has consisted of an electronic head protector and chest protector to fully automate the scoring system that enables Taekwondo athletes to deliver kicks to their opponents’ head at a much lower force than they previously had to do to score points by head kick [22,23]. Thus, we believe that injury rates in recent international Taekwondo competitions have been reported to be lower because athletes have prioritized strategies to make light contact with their opponents’ headgear and chest protector rather than using strong kicks to score points. Furthermore, we presume that the injury rate at the 2019 WTC was the lowest compared with those of previous international competitions because athletes from many countries had enough time (up to 2 years) to prepare for competitions with new strategies to prioritize light contact for scoring since the new PSS system was applied in 2017.

### 4.2. Location and Cause of Injury Based on Diagnosis in WTC

The most frequently injured area in the 2019 WTC was the lower extremity (n = 26, 43.33%). It was reported that the lower extremity was one of the major areas of injury at previous Taekwondo competitions due to the characteristic of Taekwondo, in which a point is obtained by kicking the opponent [24,25]. In Taekwondo, various styles of kicks can be used, such as front kick, turning round kick, roundhouse kick, side kick, and jumping back kick, which generates a strong force with peak linear velocities of 5.20–26.30 m/s within as short as 0.25–0.90 s [26,27]. Thus, a strong impact is delivered to the attacking athlete’s lower extremities as well as the defending athlete’s head and body. We recommend that it is necessary to improve the ability of protective equipment to absorb impact, as the injury incidence rate of the lower extremity, especially for male athletes, who generally have stronger joint forces, was 158.2/1000 AEs [28,29].

At the 2019 WTC, the most common type of injury caused by the mechanism of contact between athletes was contusion/hematoma/bruises (n = 33, 55.0%). Among the 33 injuries, the face was the most common injury site (39.39%, n = 11), followed by the neck (n = 2). Many injuries have been reported to occur above the chest protector area, such as the head, face, and neck, in previous Taekwondo competitions [20,24]. In particular, many severe injuries, such as concussion caused by a powerful impact from the kick of the opponent, have been observed [20,24,30]. However, we noticed a change in the injury profile since 2017 in the Muju WTC, where a newly improved PSS system was adopted [3]. In the 2017 Muju WTC, contusion/hematoma/bruises were the most frequently occurring injuries (n = 44, 33.6% of all injuries). The injury profiles of the 2017 and 2019 WTCs were similar as a result of the competition strategy of athletes to gain scores by making slight contact with the headgear sensor with their feet, rather than kicking, which can transfer powerful impact [22]. Apart from factors such as the fun and performance of Taekwondo competitions, the improvement of competition rules, protective equipment, and the PSS implemented by World Taekwondo can be evaluated as quite positive in terms of reducing injury incidence and concussions [22,30].

### 4.3. Injury Severity in WTC

Of all the injuries, 55.0% (n = 33) were estimated to result in an absence from competition or training of 1–3 days; 21.67% (n = 13), of 4–7 days; 15.0% (n = 9), of >7 days; and 8.33% (n = 5), did not experience time loss. Of all the injuries requiring > 4 days of time loss, 36.67% (n = 22) were classified as severe. Severe injuries included ligament ruptures/sprains (n = 7), fractures (n = 6), concussions (n = 3), dislocations/subluxation (n = 3), and muscle and tendon ruptures/strains/tendinosis (n = 3), from which athletes were not able to recover well enough to resume competitions despite on-site treatment. At the 2017 WTC, 65.6% of cases were estimated to cause time loss from competitions/training of 1–3 days; 17.6%, of 4–7 days; and 30.5%, of >7 days. Injury incidence from these two WTCs has a higher level of severity compared with other Olympic sports [13]. However, we noted that the incidence rate of severe injuries has declined only for Taekwondo. According to the injury profile of elite Korean Taekwondo athletes in 2016, 7.1% of all injuries accounted for an absence from competitions/training of 1–3 days; 14.3%, for an absence of 4–7 days; and 64.3%, of >7 days [31]. We assume that the overall IR and rate of severe injuries have declined because many athletes have changed their style of play to score and win using more accurate attacks rather than powerful kicks. Consequently, we believe that the newly revised competition rules, protective equipment, and development of the medical system have contributed to a reduction in serious injuries in international Taekwondo competitions.

### 4.4. Injury Mechanisms and Causes at WTC

At the Manchester 2019 WTC, the most common mechanism of injury observed during competitions was contact with another athlete (male athletes: 90.20%, female athletes: 68.40%), which was remarkably higher than non-contact injuries (male athletes: 7.30%; female athletes: 15.80%) and overuse injuries (male athletes: 2.40%, female athletes: 15.80%). Compared to the mechanisms of injury at the 2017 WTC, injuries due to contact with other athletes increased in both male and female athletes. In particular, male athletes sustained injuries mainly by contact with opponents, which is probably a result of differences in the power of kicks or punches generated according to sex and weight class. Furthermore, male athletes’ weight classes were higher than those of female athletes, and more male athletes belonged to heavier categories, which means male athletes are more prone to injuries by contact. In Taekwondo, a combat sport, most injuries still occur during offensive and defensive moves. Therefore, to prevent contact-type injuries, the amount of impact that protective equipment absorbs and the actual impact generated by kicks and punches must be measured and compared to develop more shock-absorbent protective equipment [29,32]. Appropriate blocking techniques and avoidance skills must be emphasized by athletes and coaches as preventive strategies during Taekwondo technical training. In addition, although non-contact injuries are less frequent than contact injuries, we still need to pay attention to non-contact mechanisms, especially for female athletes, who show a higher incidence of non-contact injuries. In future research, it will be necessary to investigate whether a Taekwondo training program with neuromuscular and proprioceptive training can reduce non-contact injuries in international competitions.

### 4.5. Study Strengths and Limitations

In our study, we analyzed the injury incidence rate, area, cause, and mechanism of injuries at the 2019 WTC, in which new competition rules and medical systems were applied. We believe that we can collect data more efficiently and accurately at larger-scale international Taekwondo competitions if athletes, coaches, physicians, registered nurses, and athletic trainers utilize the injury surveillance system (ISS) that was used in this study and also a fast and convenient medical informatics system, as the data from these can be the basis for the development of injury prevention projects. However, this study has several limitations. Taekwondo players with pre-existing injuries are highly likely to participate in international competitions before they have fully recovered because they have to continue high-intensity drills and practice sparring sessions before the competition. In addition, pre-existing injuries may affect the occurrence of other injuries during training/competition. Thus, it is difficult to determine whether the injury data collected in our study were from injuries that actually occurred during the competition. These characteristics can be more prominent in amateur/community-level competitions that do not have ISSs and medical staff.

## 5. Conclusions

Our study showed that the newly established WT competition rules and PSS at the 2019 WTC decreased the musculoskeletal injury incidence rate and number of severe injuries compared to those of previous competitions. However, the incidence rate of injuries caused by contact during an opponent’s attack is still very high. In addition, male Taekwondo players are more prone to injuries than are female players. Based on the results of this study, we strongly recommend refining Taekwondo competition rules, improving on-site venue medical systems, and developing strategic plans and safety measures for injury prevention.

## Figures and Tables

**Figure 1 ijerph-20-01978-f001:**
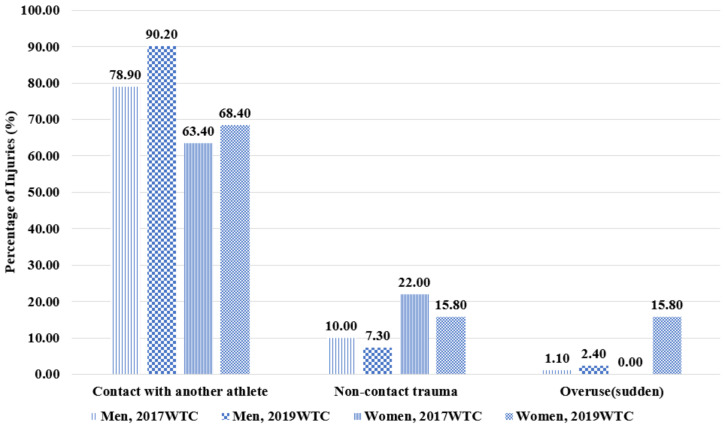
Injury Mechanisms at the 2017 and 2019 World Taekwondo Championships.

**Table 1 ijerph-20-01978-t001:** Injury rates at the 2019 World Taekwondo Championships in Manchester, United Kingdom.

Variables	Male Athletes	Female Athletes	Total
Participating athletes, No.	544	392	936
Athlete-days, No.	2720	1960	4680
Athlete-exposures, No.	1066	748	1814
Minute-exposures, min	4254	3610	7864
Injuries in competition, No.	41	19	60
Injury incidence (95% CI)			
Clinical incidence/100 athletes	7.54 (5.23, 9.84)	4.85 (2.67, 7.03)	6.41 (4.79, 8.03)
Rate/1000 athlete-days	15.07 (10.46, 19.69)	9.69 (5.33, 14.05)	12.82 (9.58, 16.06)
Rate/1000 AEs	38.46 (26.69, 50.23)	25.40 (13.98, 36.82)	33.08 (24.71, 41.45)
Rate/1000 MEs	9.64 (6.69, 12.59)	5.26 (2.90, 7.63)	7.63 (5.70, 9.56)
Incidence rate ratio/1000 AEs	1.57 (0.89, 2.76)	1	NA

Note: AEs, athlete exposure; MEs, minute exposure; 95% CI, 95% confidence interval; NA, not applicable.

**Table 2 ijerph-20-01978-t002:** Diagnosis by Injury Site at the 2019 World Taekwondo Championships in Manchester, United Kingdom.

Diagnosis	Males (% of Total)	Injury Site (No.)	Females (% of Total)	Injury Site (No.)	Total	Injury Site (No.)
Contusion/hematoma/bruise	25 (61.0)	Face (11), groin (4), thigh (3), lower leg (2), hip (1R), knee (1), ribs (1), hand (1), finger (1)	8 (42.11)	Neck (2), ribs (2), knee (1), ankle (1), foot/toe (1), hand (1)	33 (55.0)	Face (11), groin (4), thigh (3), ribs (3), neck (2), hand (2), lower leg (2), knee (2), hip (1R), finger (1), ankle (1), foot/toe (1)
Ligamentous rupture/ sprain	3 (7.32)	Knee (2NC), ankle (1NC)	4 (21.05)	Ankle (2NC), knee (1NC), wrist (1)	7 (11.67)	Knee (3NC), ankle (3NC), wrist (1)
Fracture	5 (12.20)	Hand (4), face (1)	1 (5.26)	Hand (1C)	6 (10.0)	Hand (5), face (1)
Concussion	3 (7.32)	Head/brain (3)	0	NA	3 (5.0)	Head/brain (3)
Dislocation/ subluxation	1 (2.44)	Face (1)	2 (10.53)	Shoulder (1NC, 1R)	3 (5.0)	Shoulder (1NC, 1R), face (1)
Muscle-tendon rupture/strain/ tendinosis	0	NA	3 (15.79)	Thigh (1, 1R), hip (1)	3 (5.0)	Thigh (1, 1R), hip (1)
Lesion of meniscus	0	NA	1 (5.26)	Knee (1R)	1 (1.67)	Knee (1R)
Laceration/abrasion/ skin lesion	1 (2.44)	Face (1)	0	NA	1 (1.67)	Face (1)
Muscle cramps/spasm	1 (2.44)	Thigh (1)	0	NA	1 (1.67)	Thigh (1)
Other	2 (4.88)	Face (1), groin (1)	0	NA	2 (3.33)	Face (1), groin (1)
Total	41 (100.0)	NA	19 (100.0)	NA	60 (100.0)	NA

Note: NA, not applicable; NC, non-contact; R, recurrent previous injury. The mechanism of injury for an injury site that only contains the number (no.) is contact injury. This is presented in a similar format to the table from 2017 WTC MUJU [3], for easier comparison by the readers.

## Data Availability

Not applicable.

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
