# Peer review of "Incidence of Sport Injuries in the Manchester 2019 World Taekwondo Championships: A Prospective Study of 936 Athletes from 145 Countries"

_ijerph, 2023, doi:10.3390/ijerph20031978_

Round 1
Reviewer 1 Report
This study examines the type, number and reasons for injuries in athletes competing taekwando matches. This study looked at injuries at 2 different meets that were held 2 years apart. The authors report that changes in scoring equipment and various rules resulted in a decline in the number of injuries. Although the paper is well written, there are a few typos and there are a few points in the methods and results that need to be clarified.
1) The authors do describe how some of the variables were calculated in the statistic portion of the paper. However, it would be helpful if the author actually listed the measures they collected in the methods section (and if an explanation is needed to describe how the variable were collected, this should also be in the methods section).
2) were there multiple injuries in the same athlete and if so, was there any ways to determine if one injury affected the incidence of another injury
3) the authors may want to plot incidence rates/1000 or some other measure because based on the graph it looks like there were more injuries in 2019 than 2017, but the text says there were fewer.
4) The authors report performing t-tests, but all of the data presented are descriptive and there weren't any statistics reported.
5) line 261, please change "did" to "were" to keep consistent
6) line 324 please change "reduced" to "declined"
4) why do the authors think injury was more prevalent in males than in females?
Author Response
We sincerely thank you for proceeding with the review of this manuscript. We responded as a Word file in response to the comments you provided. Modifications are marked in yellow highlight, so that existing contents and changes can be better conveyed.

Reviewer 2 Report
Well written manuscript and comprehensive study.
Interesting findings to demonstrate the possible effectiveness of rule changes and PPE in the Taekwondo, which is positive.
As well as a discussion of the comparison of the injury data across years and the possible impacts of the rule changes/PPE, etc, given the comprehensiveness of the surveillance system used in the study, perhaps it would be valuable to add a statement highlighting any possible limitations to the surveillance system at amateur level competitions. Likewise, have rule changes and PPE been promoted/utilized at amateur level comps, therefore can a comment be made about any possible reductions to injury at this level (as per manuscript conclusions)? If not, perhaps this is a recommendation of the study for future research.
There are some minor suggestions for changes throughout the manuscript (see attached file).

Author Response

(The authors gave the same response as above.)
